# COVID-19 knowledge, attitudes, and practices of United Arab Emirates medical and health sciences students: A cross sectional study

**Noura Baniyas[1], Mohamud Sheek-Hussein[2]\*, Nouf Al Kaabi[1], Maitha Al Shamsi[1], Maitha Al Neyadi[1], Rauda Al Khoori[1], Suad Ajab[2], Muhammad Abid[2], Michal Grivna[2], Fikri M. Abu-Zidan[3]**

**1** United Arab Emirates University, Al-Ain, United Arab Emirates, **2** Institute of Public Health, United Arab Emirates University, Al-Ain, United Arab Emirates, **3** Department of Surgery, College of Medicine and Health Sciences, UAE University, Al-Ain, United Arab Emirates

\* msheekhussein@uaeu.ac.ae

## Abstract

The COVID-19 pandemic is the largest viral pandemic of the 21st century. We aimed to study COVID-19 knowledge, attitudes, and practices (KAP) among medical and health sciences students in the United Arab Emirates (UAE). We performed a cross-sectional study between 2 June and 19 August 2020. The survey was distributed online using Survey Monkey. It was conducted in English and comprised two parts: socio-demographic characteristics, and KAP towards COVID-19. 712 responses to the questionnaire were collected. 90% of respondents (n = 695) were undergraduate students, while 10% (n = 81) were postgraduates. The majority (87%, n = 647) stated that they obtained COVID-19 information from multiple reliable sources. They were highly knowledgeable about the COVID-19 pandemic, but 76% (n = 539) did not recognize its routes of transmission. Medical students were significantly more knowledgeable compared with allied health students (P<0.0001, Mann Whitney U test) but there was no difference in knowledge between undergraduate and postgraduate students (P = 0.14, Mann Whitney U test). Medical students thought that more could be done to mitigate the COVID-19 situation compared with the allied health students (66.2% compared with 51.6%, p = 0.002 Fisher's Exact test). 63% (n = 431) were worried about getting COVID-19 infection, while 92% (n = 633)) were worried that a family member could be infected with the virus. 97% (n = 655) took precautions when accepting home deliveries, 94% (n = 637) had been washing their hands more frequently, and 95% (n = 643) had been wearing face masks. In conclusion, medical and health sciences students in the UAE showed high levels of knowledge and good attitudes and practices towards the COVID-19 pandemic. Nevertheless, they were worried about themselves or their family members becoming infected. Medical students had more knowledge about COVID-19 pandemic which was reflected in their opinion that more can be done to mitigate its effects.

**Data Availability Statement:** All relevant data are within the paper and its Supporting Information files.

**Funding:** None.

**Competing interests:** The authors declare that they have no competing interests.

**Abbreviations:** SARS-CoV-2, Severe acute respiratory syndrome coronavirus 2; COVID-19, Coronavirus disease 2019; HCW, Healthcare workers; ICU, Intensive care unit; ILO, International Labour Organization; IFAD, The International Fund for Agricultural Development; WHO, World Health Organization; FAO, Food and Agriculture Organization; KAP, knowledge, attitudes, and practices; UAE, United Arab Emirates; UAEU, United Arab Emirates University.

## Introduction

COVID-19 pandemic is one of the major global threats of the 21st century. Its virus (SARS-CoV-2) was isolated after causing a cluster of fatal pneumonia cases in Wuhan, China [1]. The pandemic spread swiftly because of the rapid transportation methods with major global impact on physical and mental health and on the economy [2–4]. Currently more than 150 million infected persons and 3 million deaths are attributed to this pandemic [5, 6].

The United Arab Emirates (UAE) was the first country in the Gulf Cooperation Council to report a COVID-19 case (on January 29, 2020) which was linked to Wuhan [7]. In the UAE, more than 450,000 infections and 1,470 deaths had been reported to March 28, 2021 [8]. Due to the initial lack of availability of vaccines at the outset [9], each country adopted various responses to COVID-19 to slow down transmission and to prevent oversaturation of health-care systems. The UAE government issued a set of guidelines and preventive measures to fight the spread of COVID-19, including the closure of borders, educational institutions, and shopping malls; introduction of remote working rules; restriction of public movement; and implementation of personal hygiene measures including using face masks, social distancing, mass screening of asymptomatic cases, and contact tracing [10].

If these measures are to be effective, adherence is essential, and this is influenced by people's knowledge, attitude, and practices (KAP) towards COVID-19 [11, 12]. KAP towards infectious diseases, including acceptance of immunization, are associated with emotional reactions which can affect virus transmission and control [9, 13, 14]. KAP involves a range of beliefs about the causes of the disease, risk factors, identification of symptoms, and available methods of treatment and their consequences [15]. These beliefs come from different sources, including preconceptions concerning similar viral diseases, governmental information, social media and the internet, previous personal experiences, and medical sources. These beliefs may drive preventive behaviours that can vary across different populations. Lack of knowledge or false medical beliefs may carry potential risks [16]. A study from Henan, China, showed that higher levels of information were associated with more positive attitudes towards COVID-19 preventive practices [16]. Perception of risk is important for prevention of infection during pandemics [17–20].

University students can be a source of increased health awareness and health education not only for themselves but also for those around them as they take part in the dissemination of pandemic-related knowledge supporting the prevention and control of the pandemic [21, 22]. Recent studies from Pakistan, Saudi Arabia and Japan have shown that medical students have sufficient knowledge, positive attitudes, and proactive practices during the COVID-19 crisis [23–25].

There are more than 138,000 students in the UAE colleges and universities [26]. From March 2020 their classes were shifted to online learning which was a new learning experience for them [27]. The students received information regarding COVID-19 by online lectures, webinars, university websites, LinkedIn, WhatsApp groups, Facebook pages and Newsletters. A recent study conducted in Sharjah University, UAE, showed that their students demonstrated adequate knowledge, possessed good attitudes, and had low-risk practices towards prevention of COVID-19 [28]. Nevertheless, this study was limited to a single university in UAE and included both medical and non-medical students. We thought it was important to study the KAP towards COVID-19 of medical and allied health sciences students in all UAE. Accordingly, we aimed to evaluate the knowledge of COVID-19, awareness of preventive behaviors, practice, and risk perception among the medical and allied health sciences students in the higher education institutions in the UAE.

## Materials and methods

### Ethical considerations

Ethical approval was obtained from the UAE University Social Research Committee [UAEU ERS_2020_6119]. Participants' data were anonymized at the point of registration. No personal identifiable data were collected.

### Study design

This is a cross-sectional study which was conducted among medical and health sciences students in the UAE between 2nd June and 19th August 2020.

### Sample size

We developed a sampling frame including the list of all medical and health sciences colleges and universities in the UAE. As we did not have an access to the information about the number of students at relevant institutions we were not able to estimate a sample size. As the method of data collection was an online-based survey, we used a non-probability sampling approach, namely convenience sampling. The study invitation and survey link were sent directly to the medical and health sciences colleges and universities in the UAE by e-mail and circulated on multiple social media outlets including WhatsApp©. Participants were encouraged to forward the link to their fellow medical and health sciences students and to post it on their social media platforms to maximize enrolment. Accordingly, we could not know the response rate. The study invitation included an introduction, a brief description of the study, and the link to the questionnaire. Respondents were grouped according to their educational institutions.

### Questionnaire design

The questionnaire was designed and developed in May 2020 based on two similar published studies [29, 30] and on our own review article which was accepted for publication on 25th March 2020 [3]. These three papers contained early knowledge about the COVID-19 pandemic. Zhong et al study was published on 15th March 2020 and contained 12 questions with true, false, do not know answers [29] while Taghrir et al study [30] was published on 1st April 2020 and contained 15 questions with true/false answers. The attitude and practice sections of Zhong et al [29] were measured using two questions each (agree, disagree, do not know or yes/no). The attitude section of Taghrir et al [30] was measured using two items having a 4-point Likert-type scale while practices were measured by 9 items using a yes/no answer.

 Our questionnaire was developed under the direct supervision of an infectious disease expert which was reviewed by another two experienced epidemiologists, one of them is a qualitative researcher. The questionnaire was then piloted among 10 participants for face and content validity. The questions were then modified, refined, rephrased, and restructured to be simpler and clearer. The details of our final questionnaire are attached as **S1 Appendix**. Since the COVID-19 pandemic is evolving quickly and hence influencing related knowledge and attitudes we decided to depend on face and content validity, as reliability testing was not feasible.

 Consent was taken from the participants after providing a brief description of the study, clarifying the voluntary nature of participation, and confirming the declaration of anonymity. The questionnaire was conducted in English, and comprised two parts: socio-demographic characteristics and KAP towards COVID-19. The KAP part consisted of 3 sections with a total of 34 questions, described below.

**Knowledge.** This section included 12 multiple choice and true/false questions which assessed the participants' knowledge about COVID-19. The items included etiology of the disease, transmission of the virus, symptoms, incubation period, diagnostic tests, treatment options, and prevention. In the knowledge section, respondents were given options to answer true, false, or don't know.

**Attitude.** This section included 6 questions which assessed the participants' attitudes towards the COVID-19 pandemic using a Likert scale. This was coded as follows: strongly disagree = 1, disagree = 2, undecided = 3, agree = 4, strongly agree = 5. Items included fear of getting infected, stigma around infected individuals, government measures, and participants' confidence in the measures.

**Practice.** This section included 16 questions which assessed the participants' practices related to COVID-19 using multiple-choice questions, yes/no questions, and a Likert scale. The items were related to practices and compliance with preventative measures and precautions implemented by the government, such as social distancing, wearing face masks, and hand washing.

## Statistical analysis

Knowledge level was calculated as follows: incorrect or uncertain responses were given a score of 0, and correct answers were given a score of 1. Choosing a correct answer along with an incorrect answer was given a score of 0.5. The total score for knowledge ranged from 0 to 26, with high scores indicating better knowledge of COVID-19. Poor practices were given a score of 0, and good practices were given a score of 1. Choosing a good practice with a bad practice was given score of 0.5. The total score for practices ranged from 0 to 25, with high scores indicating better COVID-19 practices.

Continuous data were presented as median (range) while categorical data were presented as number (%). Percentages were calculated from the actual available responses. Continuous data (age and scores) did not have a normal distribution, hence nonparametric statistical methods were used to compare different groups as such methods analyse the ranks, do not need a normal distribution, and can be used for small groups. Categorical data of two independent groups were compared using Fisher's Exact test. while continuous data of two independent groups were compared using Mann-Whitney U test [31]. We used the Statistical Package for the Social Sciences (IBM-SPSS version 26, Chicago, Il) for statistical analyses. A p value of < 0.05 was accepted as statistically significant.

## Results

A total of 712 responses to the questionnaire were collected. **Table 1** shows the detailed demography of the participants. 90% (n = 695) of respondents were undergraduates, while 10% (n = 81) were postgraduates. The majority of respondents (87%, n = 647) obtained COVID-19 information from multiple sources, 7% (n = 52) obtained it from social media, while the rest 6%, (n = 48) relied on either medical platforms, healthcare professionals, government media briefings, or university newsletters. 406 respondents (57%) attended webinars to learn more about COVID-19.

**Table 2** shows the comorbidities and COVID-19 history of the participants. 8% of the participants who were tested for COVID-19 had a positive result. 85% (n = 506) had had a family member or friend who was tested for COVID-19, of which 15% (n = 89) had a positive result.

## Knowledge

A total of 712 respondents completed the knowledge section of the survey (**Table 3**). 76% (n = 539) of participants did not recognize the correct routes of transmission of COVID-19,

**Table 1. Characteristics of respondents of the KAP survey collected between 2nd June and 19th August 2020.**

| Variables* | | Median or Number | Range or % |
|---|---|---|---|
| **Age (years)** | | 20 | (16–48) |
| **Gender** | | | |
| | Male | 108 | 14% |
| | Female | 690 | 86% |
| **Nationality** | | | |
| | UAE | 480 | 60% |
| | Non-UAE | 315 | 40% |
| **Emirate of residence** | | | |
| | Abu Dhabi | 421 | 54% |
| | Dubai | 125 | 16% |
| | Sharjah | 94 | 12% |
| | Ajman | 60 | 8% |
| | Ras Al Khaimah | 55 | 7% |
| | Fujairah | 15 | 2% |
| | Um Al Quwain | 6 | 1% |
| **Academic affiliation** | | | |
| | UAE University (CMHS) | 247 | 32% |
| | Fatima College of Health Sciences | 169 | 22% |
| | RAK Medical & Health Sciences University | 116 | 15% |
| | Sharjah University | 77 | 10% |
| | Mohammed Bin Rashid University | 48 | 6% |
| | Gulf Medical University | 46 | 6% |
| | Ajman University | 24 | 3% |
| | Other Colleges | 49 | 6% |
| **Speciality** | | | |
| | Medicine | 431 | 56% |
| | Nursing | 117 | 15% |
| | Pharmacy | 49 | 6% |
| | Physiotherapy | 45 | 6% |
| | Dental | 44 | 6% |
| | Radiology and Medical Imaging | 21 | 3% |
| | Biomedical Sciences | 20 | 3% |
| | Medical Laboratory Technology | 17 | 2% |
| | Others | 29 | 4% |

*All variables are expressed as number (%) except age which is expressed as median (range)

although the majority of respondents correctly recognized its symptoms, average incubation period, best diagnostic test, and its management (95%, 85%, 89%, 89% and 70% respectively). The majority of the respondents were aware of the COVID-19 preventative measures, including methods to reduce viral spread, isolation of positive cases, N95 mask use limited to health care workers, and the necessity of following preventative precautions among young adults and children (83%, 92%, 84%, and 87% respectively).

There was a highly significant difference in the overall scores representing knowledge on COVID-19 between medical students and allied health students (median (range) 17.5 (8.5–23) score compared with 16.5 (5–22.5), $P < 0.0001$, Mann Whitney U test). This represented a 22% difference of the mean rank which is considered as a practical difference. There was no

**Table 2. Comorbidities and COVID-19 history of the KAP survey respondents.**

| Variable | | Number | (%) |
|---|---|---|---|
| **Personal chronic condition** | | | |
| | Asthma | 28 | 37% |
| | Diabetes | 10 | 13% |
| | Hypertension | 5 | 7% |
| | Inflammatory bowel disease | 3 | 4% |
| | Migraine | 3 | 4% |
| | Polycystic ovary syndrome | 3 | 4% |
| | Others | 24 | 32% |
| **Personal history of COVID-19** | | | |
| | Tested for COVID-19 = yes | 160 | 92% |
| | Tested positive = yes | 13 | 8% |
| **Household history of COVID-19** | | | |
| | Asymptomatic | 16 | 19% |
| | Quarantined with mild symptoms | 57 | 67% |
| | Admitted to hospital with severe symptoms | 6 | 7% |
| | Admitted to ICU with severe symptoms | 3 | 4% |
| | Died | 3 | 4% |

significant difference in knowledge scores between undergraduate and postgraduate students (median (range) score 17.25 (5–23) compared with 17.5 (7.5–22.5), P = 0.14, Mann Whitney U test) (**Fig 1**).

## Attitude

A total of 686 respondents completed the attitudes section of the survey (**Table 4**). 63% (n = 431) of participants were worried about getting COVID-19 infection, while the vast majority (92%, n = 633) were worried that a family member could get infected with the virus. 67% (n = 461) of the respondents thought that infection with the virus is associated with stigma. 83% (n = 570) agreed that the current measures taken by the UAE government are effective in stopping the spread of the infection, and 89% (n = 614) were confident that the UAE will be able to stop the spread of the virus. Nevertheless, 60% (n = 288) thought that more measures could be implemented such as aggressive screening, full lockdown, further education for the public, monitoring the media, and combatting rumors. Some were opposed to the lockdown and suggested relaxing restrictions.

There were no significant differences in any of the negative attitudes between the medical and allied health students or undergraduate and postgraduate students (**Table 5**). Nevertheless, the medical students thought that more can be done to mitigate the situation compared with the allied health students (176/266 (66.2%) compared with 112/217 (51.6%), p = 0.002 Fisher's Exact test). This difference was not significant between the undergraduate and postgraduate students (261/435 (60%) compared with 27/49 (55.1%), p = 0.54 Fisher's Exact test).

## Practices

A total of 677 respondents completed the practices section of the survey (**Table 6**). 60% (n = 407) did not attend family gatherings, and did not visit shopping malls, coffee shops, industrial areas, hospitals or COVID-19 facilities for volunteering. 97% (n = 655) took precautions when accepting home deliveries, 94% (n = 637) had been washing their hands more

**Table 3. Responses to the survey on COVID-19 knowledge.**

| Statement | Correct | | Incorrect/ Uncertain | | Total* |
|---|---|---|---|---|---|
| | n | (%)* | n | (%)* | |
| COVID-19 is a new disease caused by virus SARS-CoV-2. | 570 | 80% | 142 | 20% | 712 |
| *Answer*: True | | | | | |
| Which animal is most likely to transmit this virus to human? | 633 | 89% | 79 | 11% | 712 |
| *Answer*: Bat, Pangolin, or Civet Cat | | | | | |
| SARS-CoV-2 can be transmitted between humans by the following routes? | 173 | 24% | 539 | 76% | 712 |
| *Answer*: Respiratory droplet, or surfaces | | | | | |
| Which of the following are COVID-19 symptoms? | 679 | 95% | 33 | 5% | 712 |
| *Answer*: Fever, Dry cough or with sputum, Shortness of breath, Nausea, Vomiting, diarrhea, Loss of taste or smell, Runny nose | | | | | |
| What is the average incubation period of COVID-19? | 602 | 85% | 110 | 15% | 712 |
| *Answer*: 7–14 days | | | | | |
| What is the best diagnostic test for COVID-19? | 636 | 89% | 76 | 11% | 712 |
| *Answer*: RT-PCR | | | | | |
| COVID-19 can be treated by using the following: | 500 | 70% | 212 | 30% | 712 |
| *Answer*: Antiviral, Anti-malarial, or Convalescent plasma transfusion | | | | | |
| Which of the following can reduce the spread of COVID-19? | 591 | 83% | 121 | 17% | 712 |
| *Answer*: Social distancing, Self-isolation, Wearing face masks, or Avoiding crowded places | | | | | |
| People who are asymptomatic and COVID-19 test positive must stay at home until they are free of the infection: | 653 | 92% | 59 | 8% | 712 |
| *Answer*: True | | | | | |
| Who should wear N95 masks? | 601 | 84% | 111 | 16% | 712 |
| *Answer*: Healthcare professionals who are dealing with COVID-19 patients | | | | | |
| Persons with COVID-19 cannot transmit the virus to others when a fever is not present. | 635 | 89% | 77 | 11% | 712 |
| *Answer*: False | | | | | |
| It is not necessary for children and young adults to take measures to prevent infection from COVID-19 virus. | 616 | 87% | 96 | 13% | 712 |
| *Answer*: False | | | | | |

Percentages may not total 100 because of rounding

*Percentages were calculated out of 712, which is the total number of respondents who completed the knowledge items

frequently, and 95% (n = 643) had been wearing face masks. Meanwhile, out of 666 respondents almost all followed curfew timings set by the UAE government (99% (n = 658)). Overall, most medical students and allied health sciences students followed proper practices.

There was no significant difference in the COVID-19 practice scores between medical students and allied health students (median (range) 15 (0–25) score of compared with 15 (0–25), P = 0.15, Mann Whitney U test, and between undergraduate and postgraduate students (median (range) score of 15 (0–25) compared with 15.5 (0–24), P = 0.3, Mann Whitney U test) (**Fig 2**).

## Discussion

Our study has shown that the majority of medical and allied health students at UAE were knowledgeable about COVID 19, worried about getting infected or having a member of their family infected, and had proper practices and precautionary measures for preventing COVID-19. Nevertheless, medical students were more knowledgeable about COVID-19 and thought that more can be done to mitigate the COVID-19 situation compared with allied health students. There was no difference in knowledge, attitudes, or practices towards COVID-19 pandemic between undergraduate and postgraduate students.

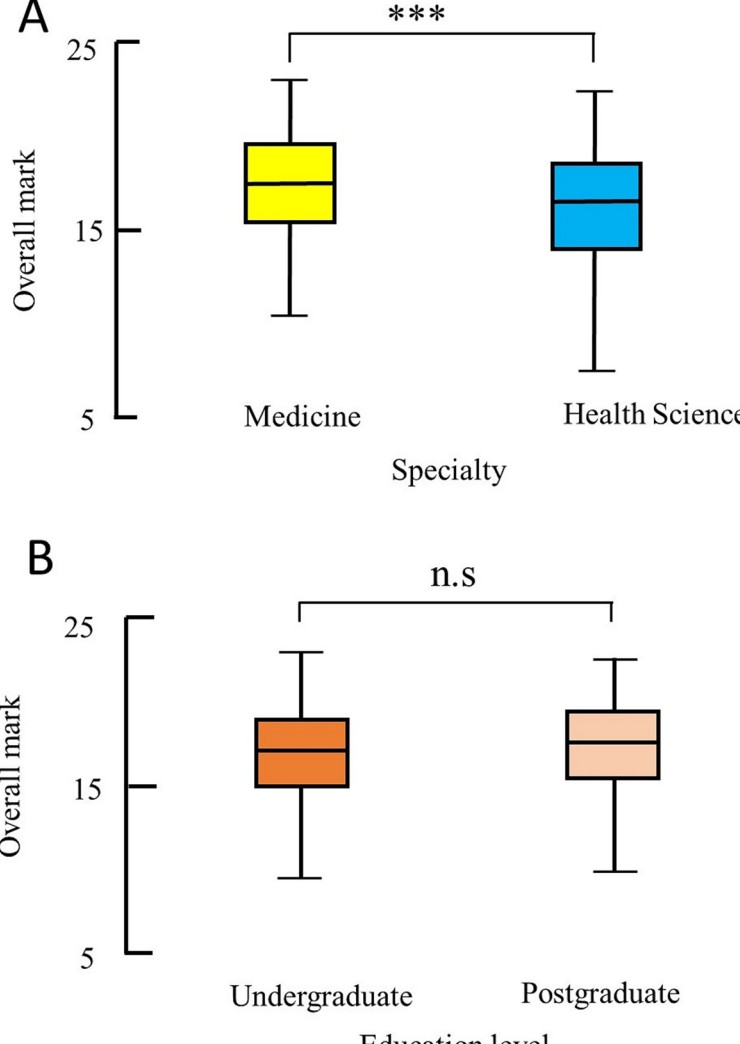

**Fig 1.** Box-and-whiskers plot of overall score for knowledge comparing medical students with allied health students (A) and undergraduate students with postgraduate students (B). The box represents the 25th to the 75th percentile IQR. The horizontal line within each box represents the median. *** = p < 0.0001, Mann-Whitney U test, ns = non significant.

A recent study of Sharjah University students [28] compared COVID19–related KAP between students of health-related and non-health related majors. Similar to our study, it was carried out during the early period of the pandemic. Students of health-related majors had higher knowledge about the COVID-19 pandemic and used face masks more compared with students of non-health related majors, despite both groups having the same attitudes [28]. A study from 10 universities in China reported that knowledge of COVID-19 was significantly higher in public universities and among medical majors compared with in private colleges and among non-medical majors [22]. In contrast other studies from Japan [25] and India [32] did not show any differences between medical and non-medical students. The high knowledge about COVID-19 among medical and allied health students in the UAE is similar to findings reported from Japan [25], Saudi Arabia [24], Portugal [33], and Pakistan [23] (96%, 86%, 82%, 72% respectively). These studies used different sets of questions to explore the KAP among

**Table 4. Responses to the survey on COVID-19 attitude.**

| Attitudinal Statement | Strongly disagree | | Disagree | | Undecided | | Agree | | Strongly agree | | Total* |
|---|---|---|---|---|---|---|---|---|---|---|---|
| **Attitude with negative feeling** | n | (%)* | n | (%)* | n | (%)* | n | (%)* | n | (%)* | |
| You are worried that you will get COVID-19 | 40 | 6% | 151 | 22% | 64 | 9% | 330 | 48% | 101 | 15% | 686 |
| You are worried that a family member can get infected with this virus | 8 | 1% | 29 | 4% | 16 | 2% | 303 | 44% | 330 | 48% | 686 |
| Infection with the virus is associated with stigma | 53 | 8% | 118 | 17% | 54 | 8% | 300 | 44% | 161 | 23% | 686 |
| **Attitude with positive feeling** | | | | | | | | | | | |
| The current measures taken by the UAE government are effective in stopping the spread of the infection | 16 | 2% | 57 | 8% | 43 | 6% | 295 | 43% | 275 | 40% | 686 |
| You are confident that the UAE will be able to stop the spread of the virus | 4 | 1% | 27 | 4% | 41 | 6% | 263 | 38% | 351 | 51% | 686 |

Percentages may not total 100 because of rounding

*Percentages were calculated out of 686, which is the total number of respondents who completed the Attitude items

students; study populations are variable; and sampling techniques are different, although most of them were carried out early in the pandemic. Studies on medical students found that they generally had good knowledge about COVID-19, although this was less in preclinical years compared with clinical years [23, 34]. Medical students are commonly asked for medical advice from their family members which prompts them to learn more about COVID-19 [35, 36].

The high level of knowledge about COVID-19 among medical and allied health students in the UAE may be attributed to their access to multiple reliable medical platforms, healthcare professionals, government media briefings, and university newsletters. These sources may have increased the existing knowledge of these students. It may also be related to the training many of them had received as volunteers in the healthcare system [37]. The majority of our respondents were aware of COVID-19 symptoms, the incubation period, diagnostic testing, management, and the preventative measures. However, only 24% in our study correctly recognized the route of COVID-19 transmission compared with other studies in which

**Table 5. Responses to the survey on COVID-19 attitude comparing medical students with allied health students and undergraduate with postgraduate student.**

| | Medical | | Allied health | | | Undergraduate | | Postgraduate | | |
|---|---|---|---|---|---|---|---|---|---|---|
| Attitude with negative feeling | Strongly agree/agree | Strongly disagree/ disagree | Strongly agree/agree | Strongly disagree/ disagree | p | Strongly agree/agree | Strongly disagree/ disagree | Strongly agree/agree | Strongly disagree/ disagree | p |
| | n (%) | disagree | n (%) | disagree | | n (%) | disagree | n (%) | disagree | |
| **Attitude with negative feeling** | | | | | | | | | | |
| You are worried that you will get COVID-19 | 239 (67.9) | 113 (32.1) | 192 (71.4) | 77 (28.6) | 0.38 | 381 (68.5) | 175 (31.5) | 50 (75.8) | 16 (24.2) | 0.26 |
| You are worried that a family member can get infected with this virus | 366 (95.6) | 17 (4.4) | 266 (93) | 20 (7) | 0.17 | 567 (94.5) | 33 (5.5) | 66 (94.3) | 4 (5.7) | 0.99 |
| Infection with the virus is associated with stigma | 259 (73) | 96 (27) | 201 (72.8) | 75 (27.2) | 0.99 | 413 (73.4) | 150 (26.6) | 48 (69.6) | 21 (30.4) | 0.57 |
| **Attitude with positive feeling** | | | | | | | | | | |
| The current measures taken by the UAE government are effective in stopping the spread of the infection | 323 (88) | 44 (12) | 246 (89.5) | 29 (10.5) | 0.62 | 508 (88.3) | 67 (11.7) | 62 (91.2) | 6 (8.8) | 0.69 |
| You are confident that the UAE will be able to stop the spread of the virus | 348 (96.4) | 13 (3.6) | 265 (93.6) | 18 (6.4) | 0.14 | 549 (95.1) | 28 (4.9) | 65 (95.6) | 3 (4.4) | 0.99 |

*Percentages were calculated out of 686, which is the total number of respondents who completed the Attitude items

**Table 6. Responses to the survey on COVID-19 practices.**

| Practice Statement | Good Practice | | Bad Practice | | Total |
|---|---|---|---|---|---|
| | n | (%) | n | (%) | |
| Have you visited any of the following places? | 407 | 60%* | 270 | 40%* | 677 |
| Shopping mall, Supermarkets, Family gatherings, coffee shops, industrial areas, hospitals for treatment, or COVID-19 facilities for volunteering | | | | | |
| Do you take precautions when accepting home deliveries? | 655 | 97%* | 22 | 3%* | 677 |
| **Have you been washing your hands more frequently?** | 637 | 94%* | 40 | 6%* | 677 |
| **Have you been wearing face masks?** | 643 | 95%† | 33 | 5%† | 676 |
| Do you follow the curfew timings set by the UAE government? | 658 | 99%‡‡ | 8 | 1%‡‡ | 666 |
| **Did you discuss COVID-19 with anyone since the pandemic started?** | 664 | 99.7%‡‡ | 2 | 0.3%‡‡ | 666 |

Percentages may not total 100 because of rounding

*Percentages were calculated out of 677, which is the total number of respondents who completed that practice item

†Percentages were calculated out of 676

‡‡ Percentages were calculated out of 666

undergraduate students were quite knowledgeable about the route of transmission [22, 23]. COVID-19 is transmitted by respiratory droplets; however, airborne transmission may be possible during a medical procedure that generates aerosols [38]. This lack of knowledge was unexpected, given that the majority of our respondents had access to freely available medical resources for this information.

The majority of our participants had followed proper practices and precautionary measures against COVID-19. 60% did not visit shopping malls, attend family gatherings, or go to coffee shops, industrial areas and hospitals; and the majority reported good preventative practices including hand washing, wearing face masks, and abiding by curfew timings. These findings could be attributed to the strict lockdown at the time when the survey was launched, access to trusted medical resources, and training in medical fields. These results are similar to those reported among undergraduate medical students in China, Pakistan, and Iran [22, 23, 31].

The participants in our study reported both negative and positive feelings regarding the pandemic, and the majority were worried that they or a member of their family might get infected. It is important to note that this study was conducted during a period of rapid increase in COVID-19 cases in the UAE. Most of the participants were confident that the UAE will be able to stop the spread of the virus. They also believed that the current measures taken by the UAE government were effective in stopping the spread of the infection although more can be done. This positive attitude can be explained by the drastic measures taken by the UAE government to contain the spread of the virus [39].

## Limitations

We have to acknowledge that our study has certain limitations. *First*, this study was conducted using an online survey and mass web-based invitation, so our response rate is unknown. The respondents were predominantly females and medical students. Consequently, the results of the questionnaire all depended on the participants self-reported behaviors, with no means of confirming whether the responses were exaggerated because of social desirability and selection bias. *Second*, this study was begun in the early stages of the pandemic, when the UAE was under lockdown, and continued for a while after the restrictions were lifted. Since then, more information about the pandemic has been published and public health measures in the UAE have changed. Thus, the results of the study may not represent the current COVID-19 KAP of

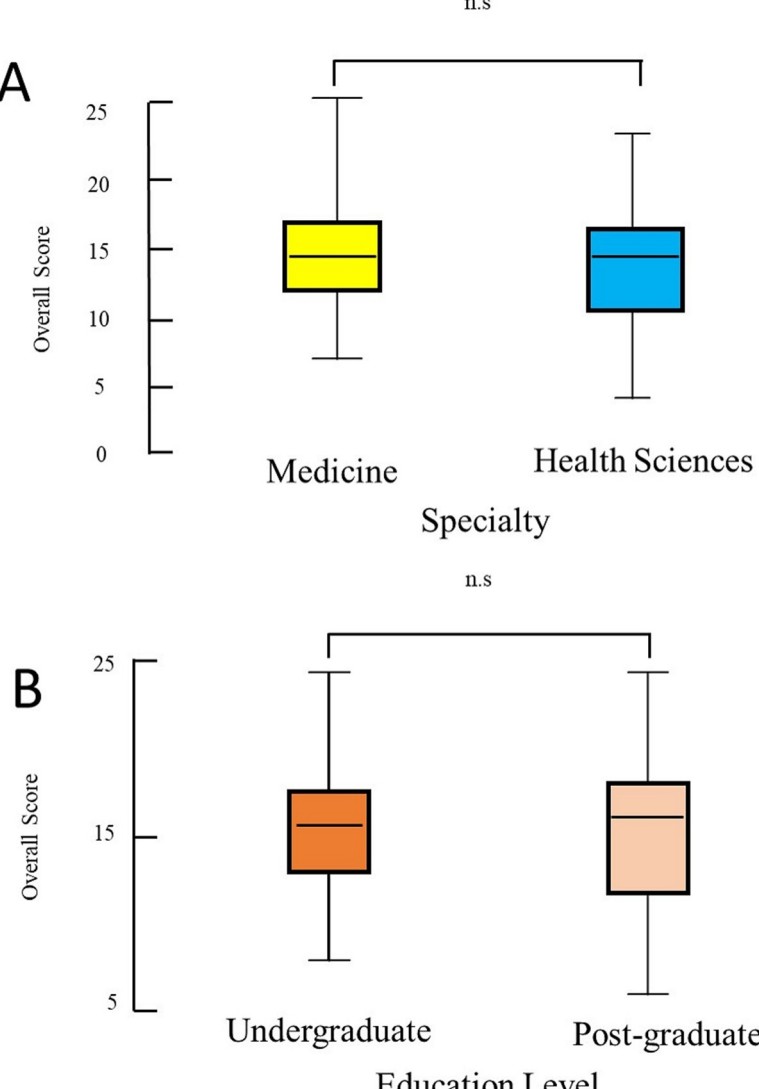

**Fig 2.** Box-and-whiskers plot of overall score of practices towards COVID-19 comparing medical students with allied health students (A) and undergraduate students with postgraduate students (B). The box represents the 25th to the 75th percentile IQR. The horizontal line within each box represents the median, ns = non significant.

medical and health sciences students. *Finally*, we did not test the reliability of our questionnaire, depending mainly on face and content validity. Nevertheless, research methodology in crisis situations differs from ordinary situations. We were keen to collect information to help us in our disaster planning for mitigation despite some shortcomings in the methodology [3, 4]. A disaster is defined as "a situation in which available resources are insufficient for immediate need of medical care" [40]. COVID-19 epidemic is a disaster of the highest nature [41]. Research methods and ethics during disasters must be adapted quickly to accommodate the situation so that that mitigation can be achieved, and lessons learned can be carried for the future [42, 43]. The rapidly evolving situation of the COVID-19 pandemic, with its concomitant changes in knowledge and practices, made it difficult to assess the reliability of our

questionnaire in a routine way especially in the knowledge section, as our understanding of COVID-19 was continuously changing.

## Conclusions

Medical and health sciences students in the UAE showed high levels of knowledge, good attitudes, and good practices towards the COVID-19 pandemic. Nevertheless, they were worried about infection both of themselves and their family members. Medical students were more knowledgeable about COVID-19 and thought that more can be done to mitigate the COVID-19 situation compared with allied health students.

## Supporting information

**S1 Appendix.**
(PDF)

**S1 Data.**
(XLSX)

## Acknowledgments

The authors thank Ms. Geraldine Kershaw, Lecturer, Medical Communication and Study Skills, Department of Medical Education, College of Medicine and Health Sciences, UAE University, for her professional linguistic and grammar corrections. We are also thankful to Dr. Ahmed R. Alsuwaidi, Dr. Iffat Elbarazi, Laila Masood, Ph.D. Candidate, Dr. Marilia Silva Paulo for their advice during developing this project, and for Ms. Laila Masood for facilitating the SurveyMonkey.

## Author Contributions

**Conceptualization:** Noura Baniyas, Mohamud Sheek-Hussein, Nouf Al Kaabi, Maitha Al Shamsi, Maitha Al Neyadi, Rauda Al Khoori, Muhammad Abid, Michal Grivna.

**Data curation:** Fikri M. Abu-Zidan.

**Formal analysis:** Suad Ajab.

**Methodology:** Nouf Al Kaabi, Maitha Al Shamsi, Maitha Al Neyadi, Rauda Al Khoori.

**Project administration:** Fikri M. Abu-Zidan.

**Supervision:** Mohamud Sheek-Hussein, Fikri M. Abu-Zidan.

**Validation:** Fikri M. Abu-Zidan.

**Writing – original draft:** Noura Baniyas, Mohamud Sheek-Hussein, Nouf Al Kaabi, Maitha Al Shamsi, Maitha Al Neyadi, Rauda Al Khoori, Suad Ajab, Muhammad Abid, Michal Grivna, Fikri M. Abu-Zidan.

**Writing – review & editing:** Noura Baniyas, Mohamud Sheek-Hussein, Nouf Al Kaabi, Maitha Al Shamsi, Maitha Al Neyadi, Rauda Al Khoori, Suad Ajab, Muhammad Abid, Fikri M. Abu-Zidan.

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
