## [Decision Letter · Decision Letter 0]

16 Mar 2021

PONE-D-21-01418

COVID-19 Knowledge, Attitudes, and Practices of United Arab Emirates Medical and Health Sciences Students: A Cross Sectional Study

PLOS ONE

Dear Dr. SHEEK-HUSSEIN,

Thank you for submitting your manuscript to PLOS ONE. After careful consideration, we feel that it has merit but does not fully meet PLOS ONE’s publication criteria as it currently stands. Therefore, we invite you to submit a revised version of the manuscript that addresses the points raised during the review process.

We look forward to receiving your revised manuscript.

Kind regards,

Jenny Wilkinson, PhD

Academic Editor

PLOS ONE

Journal Requirements:

4. We note you have included a table to which you do not refer in the text of your manuscript. Please ensure that you refer to Table 5 in your text; if accepted, production will need this reference to link the reader to the Table.

5. Please include additional information regarding the survey or questionnaire used in the study and ensure that you have provided sufficient details that others could replicate the analyses.

For instance, if you developed a questionnaire as part of this study and it is not under a copyright more restrictive than CC-BY, please include a copy, in both the original language and English, as Supporting Information.

Furthermore, please provide additional information regarding the validation of the questionnaire.

Additional Editor Comments:

Thank you for your submission, two reviewers’ comments are attached. As one of the reviewers provided very brief comments, I have provided additional comments.

1. While the language in the work is generally acceptable it would benefit from careful proof reading to ensure correct grammar and sentence structure is used

2. There is a large and growing body of literature on COVID impacts on students and their attitudes etc, this is not reflected in the Introduction to your work and the work would be strengthen by a more comprehensive review of literature on this topic

3. Please provide more detailed information about the construction and testing of the survey instrument, for example what is the validity and reliability of the instrument?

4. The statistical analysis is rather superficial as it is presented as basis descriptive statistics – the work would be strengthened by more comprehensive analysis

5. In Table 1 a number of institutions are listed however it is unclear what type of sampling method was used to ensure that the sample was representative

6. The column titles are incorrect for the age item in Table 1

7. In Table 3 where questions have multiple answers these should be provided so readers know what is selected as correct or incorrect

8. With strengthening and improvements to the Introduction and analysis this will allow the Discussion to be recrafted to provide a more comprehensive and critical discussion of the findings

Reviewers' comments:

Reviewer's Responses to Questions

**Comments to the Author**

1. Is the manuscript technically sound, and do the data support the conclusions?

Reviewer #1: Partly

Reviewer #2: Partly

2. Has the statistical analysis been performed appropriately and rigorously? 

Reviewer #1: I Don't Know

Reviewer #2: Yes

3. Have the authors made all data underlying the findings in their manuscript fully available?

Reviewer #1: Yes

Reviewer #2: No

4. Is the manuscript presented in an intelligible fashion and written in standard English?

Reviewer #1: No

Reviewer #2: No

5. Review Comments to the Author

Reviewer #1: The research topic and methodology is similar to research entitled: A Cross-Sectional Study on University Students’ Knowledge, Attitudes, and Practices Toward COVID-19 in the United Arab Emirates, Am. J. Trop. Med. Hyg., 104(1), 2021, pp. 75–84, doi:10.4269/ajtmh.20-0857. I think this research don have primary conditions for publishing.

Reviewer #2: Dear authors,

Please consider the following issues:

1- In the keywords section, you should use MeSH terms. KAP is not a suitable term. It is not only an abbreviation, but also a non-MeSH term. Additionally, UAE should be changed. Check out MeSH database to use the correct form of the term.

2- The running title is not suitable and it should be revised.

3- Introduction is not well-developed. The rationale and necessities of conducting this research is not enough described. Also, more articles are needed in this section.

4- “The estimated population of UAE is 9,948,495”. Why did you state this information?! It is not necessary! You can state the reasons. The sample size section should be revised comprehensively.

5- Information about validity and reliability of the tool should be offered.

6- Statistical analysis section should be stated in more details.

7- State response rate in the results.

8- The style of writing in some parts is not appropriate and not written academically and professionally. Please read and check out the whole article for these issues (e.g. line 136 page 8). Mind grammatical, and dictation issues too.

9- Some parts of the discussion are similar to results or suit there. This part should be more narrative and contain more comparative information based on more similar studies. It is not common to state numerical data in this part.

6. PLOS authors have the option to publish the peer review history of their article (what does this mean?). If published, this will include your full peer review and any attached files.

Reviewer #1: **Yes: **Yes

Reviewer #2: No

---

## [Author Response · Author response to Decision Letter 0]

4 Apr 2021

Dear Professor Wilkinson

Thank you for your prompt response and giving us the chance to re-submit a revised version of the above manuscript. The manuscript has now been completely re-written as advised by the reviewers. We thank you and the reviewers for their highly encouraging and constructive comments, which have significantly improved our manuscript. All changes made in the manuscript are highlighted in yellow color to facilitate the review process. The answers to the Editorial and reviewers’ comment are as follows:

Journal Requirements

Comment 1. Please ensure that your manuscript meets PLOS ONE's style requirements, including those for file naming. 

Answer: We confirm that our manuscript meets PLOS ONE's style requirements.

Comment 2. We note that you have stated that you will provide repository information for your data at acceptance. Should your manuscript be accepted for publication, we will hold it until you provide the relevant accession numbers or DOIs necessary to access your data. 

Answer: The information is submitted as an Excel supplementary file which can be published when the paper is accepted.

Comment 3. Your ethics statement should only appear in the Methods section of your manuscript. If your ethics statement is written in any section besides the Methods, please delete it from any other section.

Answer: We confirm that the ethics statement is written only in the Methods section.

Comment 4. We note you have included a table to which you do not refer in the text of your manuscript. Please ensure that you refer to Table 5 in your text; if accepted, production will need this reference to link the reader to the Table.

Answer: Table 5 (now Table 6) has now been cited in the manuscript (Page 16).

Comment 5. Please include additional information regarding the survey or questionnaire used in the study and ensure that you have provided sufficient details that others could replicate the analyses. For instance, if you developed a questionnaire as part of this study and it is not under a copyright more restrictive than CC-BY, please include a copy, in both the original language and English, as Supporting Information.

Answer: The survey is now attached as a supplementary file (Appendix 1, Page 6, Paragraph 3) and more details about its development are added as requested (Page 6, Paragraph 2).

Comment 6: Furthermore, please provide additional information regarding the validation of the questionnaire.

Answer: This has now been answered in comment 3 of the Editor and discussed in the limitations of the study (Page 6, Paragraph 2; Page 21, last paragraph).

Additional Editor Comments

Thank you for your submission, two reviewers’ comments are attached. As one of the reviewers provided very brief comments, I have provided additional comments.

Comment 1. While the language in the work is generally acceptable it would benefit from careful proof reading to ensure correct grammar and sentence structure is used

Answer: Thank you for your advice. The manuscript has now been edited by a professional English language Lecturer. Kindly note the acknowledgment section (Page 24) and attached certificate of linguistic editing. 

Comment 2. There is a large and growing body of literature on COVID impacts on students and their attitudes etc, this is not reflected in the Introduction to your work and the work would be strengthen by a more comprehensive review of literature on this topic.

Answer: Thank you for your kind advice. The introduction has now been modified as suggested by performing a comprehensive review of the literature (Pages 3 and 4).

Comment 3. Please provide more detailed information about the construction and testing of the survey instrument, for example what is the validity and reliability of the instrument?

Answer: Thank you for your important comment. We have now provided more information about the construction and testing of the survey instrument as requested and added it as an appendix (Page 6, Paragraph 2; appendix 1).

Comment 4. The statistical analysis is rather superficial as it is presented as basis descriptive statistics – the work would be strengthened by more comprehensive analysis.

Answer: Thank you for your advice which is highly appreciated. We have now quantified the knowledge and practice (methods, Page 7, Paragraph 3) and compared the results of knowledge (Page 12, last paragraph), attitudes (Page 15) and practices (Page 18) between medical students and allied health students and between undergraduate with postgraduate students as requested. This resulted in an extra table (Table 5, Pages 15 and 16); and two new figures (Figures 1, Page 13; and Figure 2, Page 18).

Comment 5. In Table 1 a number of institutions are listed however it is unclear what type of sampling method was used to ensure that the sample was representative

Answer: Thank you for your comment. As we did not have an access to the information about the number of students at relevant institutions we were not able to estimate a sample size. As the method of data collection was an online-based survey, we used a non-probability sampling approach, namely convenience sampling. This has now been clarified (Page 5, Paragraph 3).

Comment 6. The column titles are incorrect for the age item in Table 1

Answer: The table has been corrected as advised. All variables are expressed as number (%) except age which is expressed as median (range); (Page 8, Page 9 footnote)

Comment 7. In Table 3 where questions have multiple answers these should be provided so readers know what is selected as correct or incorrect

Answer: The correct answers have been incorporated in the Table 3 as recommended (Pages 11 and 12).

Comment 8. With strengthening and improvements to the Introduction and analysis this will allow the Discussion to be recrafted to provide a more comprehensive and critical discussion of the findings

Answer: The discussion has been modified as advised to be more comprehensive and critical to our findings (Page 19-21).

Reviewers' comments

Reviewer 1 

Comment: The research topic and methodology is similar to research entitled: A Cross-Sectional Study on University Students’ Knowledge, Attitudes, and Practices Toward COVID-19 in the United Arab Emirates, Am. J. Trop. Med. Hyg., 104(1), 2021, pp. 75–84, doi:10.4269/ajtmh.20-0857. I think this research don have primary conditions for publishing.

Answer: Thank you for your very useful comment which is highly appreciated. We have now critically read the mentioned paper (reference 28 in our paper). It covers only Sharjah University students and aimed to compare COVID19–related KAP between health-related (HR) and non-HR (NHR) majors. It was done during the first two weeks of May 2020 (early of the pandemic) and published end of November 2020 while ours targeted all UAE Medical and Health Sciences students and was run between 2nd June and 19th August 2020 (2 months). It is very common that researchers think of the same important question at the same time. We regret that were not aware of this study once we submitted our paper. We have now included it in the introduction (Page 4, last Paragraph) and discussion (Page 19, Second Paragraph).

Reviewer 2 

Please consider the following issues:

Comment 1: In the keywords section, you should use MeSH terms. KAP is not a suitable term. It is not only an abbreviation, but also a non-MeSH term. Additionally, UAE should be changed. Check out MeSH database to use the correct form of the term.

Answer: Thank you for your advice. The key words have been chosen to be MeSH terms (Page 2). 

Comment 2: The running title is not suitable and it should be revised.

Answer: Thank you for your advice. The running title has been changed as advised (Page 1). 

Comment 3: Introduction is not well-developed. The rationale and necessities of conducting this research is not enough described. Also, more articles are needed in this section.

Answer: Thank you for your kind advice. The introduction has now been modified as suggested by performing a comprehensive review of the literature (Pages 3 and 4).

Comment 4: “The estimated population of UAE is 9,948,495”. Why did you state this information?! It is not necessary! You can state the reasons. The sample size section should be revised comprehensively.

Answer: Thank you for important concern. We have now revised the sample size section as advised (Page 5, last Paragraph). It reads now as follows: “We developed a sampling frame including the list of all medical and health sciences colleges and universities in the UAE. As we did not have an access to the information about the number of students at relevant institutions we were not able to estimate a sample size. As the method of data collection was an online-based survey, we used a non-probability sampling approach, namely convenience sampling”. 

Comment 5: Information about validity and reliability of the tool should be offered.

Answer: We have now provided more information about the validity and reliability of the study tool (Page 6, Paragraph 2) and discussed it in the limitations of the study (Page 21, last Paragraph).

Comment 6: Statistical analysis section should be stated in more details.

Answer: This section has been detailed as requested (Page 7 paragraphs 3 and 4; and Page 8 Paragraph 1). 

Comment 7: State response rate in the results.

Answer: Thank you for your concern. As the method of data collection was an online-based survey, we used a non-probability sampling approach, namely convenience sampling. Accordingly, we could not know the response rate. We have now highlighted this in the methods (Page 5, last paragraph) and discussed it in the limitations sections (Page 21, Paragraph 2).

Comment 8: The style of writing in some parts is not appropriate and not written academically and professionally. Please read and check out the whole article for these issues (e.g. line 136 page 8). Mind grammatical, and dictation issues too.

Answer: Thank you for your advice. The manuscript has now been edited by a professional English language Lecturer. Kindly note the acknowledgment section (Page 24) and attached certificate of linguistic editing. 

Comment 9: Some parts of the discussion are similar to results or suit there. This part should be more narrative and contain more comparative information based on more similar studies. It is not common to state numerical data in this part.

Answer: Thank you for you kind advice. We have revised the discussion taking into consideration these points (Pages 19-21).

Thank you for your consideration for this revised manuscript. We hope that these changes will satisfy the reviewers and that the manuscript finally finds a place in your reputable journal.

Yours Sincerely

Mohamud Sheek-Hussein

---

## [Decision Letter · Decision Letter 1]

19 Apr 2021

PONE-D-21-01418R1

COVID-19 Knowledge, Attitudes, and Practices of United Arab Emirates Medical and Health Sciences Students: A Cross Sectional Study

PLOS ONE

Dear Dr. SHEEK-HUSSEIN,

Thank you for submitting your manuscript to PLOS ONE. After careful consideration, we feel that it has merit but does not fully meet PLOS ONE’s publication criteria as it currently stands. Therefore, we invite you to submit a revised version of the manuscript that addresses the points raised during the review process.

We look forward to receiving your revised manuscript.

Kind regards,

Jenny Wilkinson, PhD

Academic Editor

PLOS ONE

Journal Requirements:

Additional Editor Comments (if provided):

Thank you for your responses to reviewer comments and modifications to the manuscript.

1. On lines 156-158 rather than using the term ‘bad practices’ it is suggested that ‘poor practices’ is used

2. In figure 1 there is a statistical difference shown in A however the practical difference appears small; please comment on whether this is a practical difference. Please also double check that this is in fact a statistically significant difference.

Please also consider the comments of Reviewer 2 and determine what other changes could be made to improve the work.

Reviewers' comments:

Reviewer's Responses to Questions

**Comments to the Author**

1. If the authors have adequately addressed your comments raised in a previous round of review and you feel that this manuscript is now acceptable for publication, you may indicate that here to bypass the “Comments to the Author” section, enter your conflict of interest statement in the “Confidential to Editor” section, and submit your "Accept" recommendation.

Reviewer #1: All comments have been addressed

Reviewer #2: (No Response)

2. Is the manuscript technically sound, and do the data support the conclusions?

Reviewer #1: Yes

Reviewer #2: Partly

3. Has the statistical analysis been performed appropriately and rigorously? 

Reviewer #1: Yes

Reviewer #2: Yes

4. Have the authors made all data underlying the findings in their manuscript fully available?

Reviewer #1: Yes

Reviewer #2: Yes

5. Is the manuscript presented in an intelligible fashion and written in standard English?

Reviewer #1: No

Reviewer #2: No

6. Review Comments to the Author

Reviewer #1: All corrections have been done well. All parts like introduction, methodology, findings and discussion include details information.

Reviewer #2: Dear authors,

Some comments have not been addressed suitably including the discussion, methods, …. And the revision could not convince me for further steps.

The rationale of conducting this research was not stated clearly. What was the main necessities of doing this study? There were many similar studies in this field and the results may be predictable. The previous study (ref. number 28) is very similar to this study so what was the need?

The process of questionnaire’s modification was not clearly discussed.

Additionally, the methods section is too common and repeated and there is not enough novelty in the research methodologically. Also, in my opinion, the last limitation may not be acceptable.

7. PLOS authors have the option to publish the peer review history of their article (what does this mean?). If published, this will include your full peer review and any attached files.

Reviewer #1: **Yes: **no

Reviewer #2: No

---

## [Author Response · Author response to Decision Letter 1]

22 Apr 2021

Dear Professor Wilkinson

Thank you for your prompt response and giving us the chance to re-submit a second revised version of the above manuscript. The manuscript has now been revised as advised. We thank you and the reviewers for their highly encouraging and constructive comments, which have significantly improved our manuscript. All changes made in the manuscript are highlighted in yellow color to facilitate the review process. The answers to the Editorial and reviewers’ comment are as follows:

Journal Requirements

Comment 1. Please review your reference list to ensure that it is complete and correct. If you have cited papers that have been retracted, please include the rationale for doing so in the manuscript text, or remove these references and replace them with relevant current references. Any changes to the reference list should be mentioned in the rebuttal letter that accompanies your revised manuscript. If you need to cite a retracted article, indicate the article’s retracted status in the References list and also include a citation and full reference for the retraction notice.

Answer: Thank you for your valuable comment. The references have now been changed following exactly your instructions. All changes made in the references are highlighted by yellow. We replaced reference 13 by another recent reference giving the same message which is easier to find in PUBMED. We have also added four needed references to the limitation section (References 40-43).

Editor’s Comments

Thank you for your responses to reviewer comments and modifications to the manuscript.

Comment 1. On lines 156-158 rather than using the term ‘bad practices’ it is suggested that ‘poor practices’ is used

Answer: Thank you for your kind advice. The term has now been corrected as requested (Page 8, line 168).

Comment 2. In figure 1 there is a statistical difference shown in A however the practical difference appears small; please comment on whether this is a practical difference. Please also double check that this is in fact a statistically significant difference.

Answer: 

Thank you for your kind concern. Please be assured that the statistical analysis is correct, and the reporting is accurate. The senior author of the current paper (FA-Z) is the Statistical Editor of World Journal of Emergency Surgery (IF=4.1) and he did the analysis and drew the graphs himself. Kindly note that the overall score was skewed as shown in the figure below and that is why we used nonparametric methods and presented the figures as Box plot and IQ range which is the proper way. Kindly note that the medians are different. The Mann Whitney test compares the ranks, and the difference of the ranks is more than 22% which we consider as a practical difference. This has now been added to the results section as requested (Page 13, line 213).

Comment 3. Please also consider the comments of Reviewer 2 and determine what other changes could be made to improve the work.

Answer: Thank you for your important advice which we have genuinely followed. We have tried our best to accommodate as much as possible from the reviewer’s comments within the manuscript. It is obvious that the reviewer recommended rejection depending on the below addressed concerns.

1. The reviewer thinks that the introduction does not lead to a clear aim. We completely disagree with this comment. The introduction has a classical reversed pyramid approach in which we started broad and then narrowed the discussion gradually to reach a well-defined aim at the end of the introduction.

2. The reviewers thinks that this study is similar the study of reference 28 which is not the case at all. Reference 28 studied a single university in UAE with both medical and non-medical students while we have studied medical and allied health students in all UAE. Our study has much more generalizability compared with reference 28. We have clarified this in the introduction (Page 4, lines 91-93). 

3. The reviewer based his rejection on the basis that the methods are not novel. We have submitted our paper to PLOS ONE because it does not consider lack of novelty as a criterion for rejection and its main objective is to overcome this publication bias stressing on publishing good research. We think that our study lies within this domain. Furthermore, methods of studies are usually used before and not novel. 

4. The reviewers asked for more details on the way the questionnaire was developed which we have now added to the methods section as requested (Page 6, lines 120-134).

5. The reviewer has the opinion that research methods should be the same in normal and disaster situations which we completely disagree with. The senior author of the current paper (F A-Z) is an international expert with extensive practical experience in disaster medicine. This point has now been more clarified and referenced in the limitation section (Page 22, lines 324- 328 and Page 23, lines 329-331). 

Finally, it will be assuring for the reviewer that his/her anonymous comments will be published alongside the publication of this paper beside our response so these comments and concerns will be shared with the general readers.

Reviewers' comments

Reviewer 1 

Comment: All corrections have been done well. All parts like introduction, methodology, findings and discussion include details information.

Answer: Thank you for your previous advice and highly encouraging comment. Nothing to answer.

Reviewer 2 

Comment 1: a) Some comments have not been addressed suitably including the discussion, methods, …. And the revision could not convince me for further steps. The rationale of conducting this research was not stated clearly. What was the main necessities of doing this study? There were many similar studies in this field and the results may be predictable. b) The previous study (ref. number 28) is very similar to this study so what was the need?

6. Answers: a) We completely disagree with this comment. The introduction has a classical reversed pyramid approach in which we started broad and then narrowed the discussion gradually to reach a well-defined aim at the end of the introduction. b) Our study is different from reference 28 which studied a single university in UAE with both medical and non-medical students while we have studied medical and allied health students in all UAE. Our study has much generalizability compared with reference 28. We have clarified this in the introduction (Page 4, lines 91-93).

Comment 2: a) The process of questionnaire’s modification was not clearly discussed. b) Additionally, the methods section is too common and repeated and there is not enough novelty in the research methodologically. 

7. Answer: Thank you for your concerns. 

a) The details of how the questionnaire was developed has now been added to the methods section as requested (Page 6, lines 120-134). 

b) We are aware of this point. We have submitted our paper to PLOS ONE because it does not consider lack of novelty as a criterion for rejection and its main objective is to overcome this publication bias stressing on publishing good research. We think that our study lies within this domain. Furthermore, methods of studies are usually used before and not novel. 

Comment 3: Also, in my opinion, the last limitation may not be acceptable.

Answer: Thank you for your expressed opinion which is well appreciated. Kindly permit us to disagree with you. The senior author of the current paper (F A-Z) is an international expert with extensive practical experience in disaster medicine. This point has now been more clarified and referenced in the limitation section (Page 22, lines 324- 328 and Page 23, lines 329-331). 

Thank you for your consideration for this revised manuscript. We hope that these changes will be adequate and answer the raised concerns, and that the manuscript finally finds a place in your reputable journal.

Yours Sincerely

Mohamud Sheek-Hussein and Fikri Abu-Zidan

---

## [Editor Report · Decision Letter 2]

26 Apr 2021

COVID-19 Knowledge, Attitudes, and Practices of United Arab Emirates Medical and Health Sciences Students: A Cross Sectional Study

PONE-D-21-01418R2

Dear Dr. SHEEK-HUSSEIN,

We’re pleased to inform you that your manuscript has been judged scientifically suitable for publication and will be formally accepted for publication once it meets all outstanding technical requirements.

Kind regards,

Jenny Wilkinson, PhD

Academic Editor

PLOS ONE

Additional Editor Comments (optional):

Thank you for responding to the reviewer and editor comments; these have satisfactorily address the issues raised
---

## [Editor Report · Acceptance letter]

3 May 2021

PONE-D-21-01418R2 

COVID-19 Knowledge, Attitudes, and Practices of United Arab Emirates Medical and Health Sciences Students: A Cross Sectional Study 

Dear Dr. Sheek-Hussein:

I'm pleased to inform you that your manuscript has been deemed suitable for publication in PLOS ONE. Congratulations! Your manuscript is now with our production department. 

Kind regards, 

on behalf of

Dr Jenny Wilkinson 

Academic Editor

PLOS ONE